# Score-based Idempotent Distillation of Diffusion Models

## Abstract

Idempotent generative networks (IGNs) are a new line of generative models based on the idea of idempotent mapping to a target manifold. IGNs support both single- and multi-step generation, allowing for a flexible trade-off between computational cost and sample quality. But similar to Generative Adversarial Networks (GANs), conventional IGNs require adversarial training and are prone to training instabilities and mode collapse. Diffusion and score-based models are popular approaches to generative modeling that iteratively transport samples from one distribution, usually a Gaussian, to a target data distribution. These models have gained popularity due to their stable training dynamics and high-fidelity generation quality. However, this stability and quality come at the cost of high computational cost, as the data must be transported incrementally along the entire trajectory. New sampling methods, model distillation, and consistency models have been developed to reduce the sampling cost and even perform one-shot sampling from diffusion models. In this work, we unite diffusion and idempotent models by training idempotent models through distillation from diffusion models' scores. Our proposed method to train IGNs is highly stable and does not require adversarial losses. We provide a theoretical analysis of our proposed score-based training methods. We empirically show that idempotent networks can be effectively distilled from a pre-trained diffusion model, enabling faster inference compared to iterative score-based models. Like IGNs and score-based models, SIGNs can perform multi-step sampling, allowing users to trade off quality for efficiency. As these models operate directly on the source domain, they can project corrupted or alternate distributions back onto the target manifold, enabling zero-shot editing of inputs. We validate our models on a simple multi-modal dataset as well as multiple image datasets, achieving state-of-the-art results for idempotent models on the CIFAR and CelebA datasets.

## 1 Introduction

Generative modeling for high-dimensional data like images and video faces a fundamental trilemma Xiao et al. (2021): balancing (a) high sample quality, (b) fast sampling, and (c) diverse mode coverage Yu et al. (2020); Zhao et al. (2018). This challenge has driven the development of numerous deep generative methods, each navigating these trade-offs differently. Prominent examples include generative adversarial networks (GANs) Goodfellow et al. (2020), variational autoencoders (VAEs) Kingma and Welling (2013), auto-regressive models Van Den Oord et al. (2016), normalizing flows Rezende and Mohamed (2015); Dinh et al. (2014); Ho et al. (2019), flow-matching, and diffusion models Sohl-Dickstein et al. (2015); Ho et al. (2020); Nichol and Dhariwal (2021).

GANs generate high-quality samples quickly but suffer from training instabilities and reduced mode coverage due to their adversarial objective, which can lead to mode collapse Salimans et al. (2016); Kodali et al. (2017); Jo et al. (2020). Also, capable of single-step sampling, normalizing flows and VAEs often produce lower-quality samples Ho et al. (2022a).

Idempotent Generative Models (IGNs) Shocher et al. (2023) are the newest entry in the zoo of generative models. They are a novel class of GAN-like models that can combine the benefits of both diffusion models and GANs. They support single-step sampling and iterative refinement, offering a flexible trade-off between computational cost and sample quality. Like GANs, IGNs suffer from training instabilities stemming from their adversarial objective. Modern generative model training

requires a large amount of data and computing resources. Models with unstable training, where the model can abruptly diverge, are prohibitively costly to train. Improving the training stability of IGNs is crucial for effective design exploration and for enhancing model performance.

Diffusion and score-matching models have become the de facto generative models. They achieve high sample quality and are significantly easier to optimize. This training stability has enabled researchers to rigorously optimize these models, leading to architectural innovations and state-of-the-art performance in domains such as image, video Ho et al. (2022b); Mei and Patel (2023), audio generation Kong et al. (2020), molecular synthesis Schneuing et al. (2022); Hoogeboom et al. (2022); Xu et al. (2022), and protein structure prediction Corso et al. (2022). However, this performance comes at the cost of slow, iterative sampling, as these models require multiple steps to transform a sample from a simple prior distribution to the complex data distribution. To address this limitation, many recent works have focused on accelerating sampling through methods like distillation, without sacrificing model quality or stability Song and Dhariwal (2023); Xiao et al. (2021); Sauer et al. (2023).

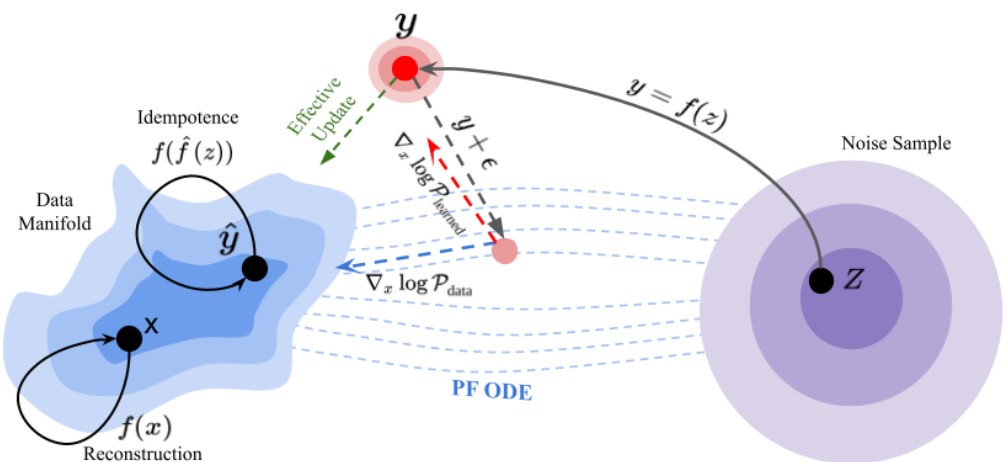

Figure 1: The goal of the optimal idempotent network, $\hat{f}$, is to project inputs outside of the target manifold on the left, $z \sim Z$, onto that manifold. While imposing dataset reconstruction $\left(\hat{f}(x) = x\right)$ and idempotence $\left(f\left(\hat{f}(z)\right) = \hat{f}(z)\right)$, our method uses estimation of the real score, $\nabla_x \log \mathcal{P}_{\text{real}}$. The projected output $y = f(Z)$ is on the data manifold when our model learned score function, $\nabla_x \log_{\text{learned}}$ is equal to $\nabla_x \log \mathcal{P}_{\text{real}}$. We design our training algorithms to estimate the real score function and train the model.

To enable exploration of IGNs on diverse domains and large-scale high-resolution datasets, we must improve the training characteristics of the optimization methods. Inspired by the success of diffusion models, we propose an optimization algorithm to stabilize IGN training. Drawing from consistency models Song et al. (2023), which learn to map points along a probability flow trajectory, we introduce Score-based Idempotent Generative Networks (SIGNs). SIGNs are trained to map noisy samples back onto the data manifold. They can be viewed as implicit time consistency models that support arbitrary noise schedules. We reformulate the IGN objective as a projection problem: noisy samples, which lie far from the data manifold (Fig. 1), are projected back onto it by the model, which remains idempotent for samples already on the manifold. This connection to score models enables the transfer of architectures, training techniques, and pre-trained weights to IGNs, while their single-step generation capabilities significantly improve sampling speed. SIGNs can be trained independently or by distilling a pre-trained diffusion model. They are capable of single-step and multi-step sampling and zero-shot editing. We establish a connection between diffusion models (or, equivalently, score-based models) and IGNs and propose an alternative objective for training IGNs efficiently.

**Contributions** In our current work, we present: **(a)** a new, stable objective combining score-matching and tightening losses to replace the unbounded, unstable tightening loss in IGNs; **(b)** a theoretical analysis of our proposed objective highlighting; and **(c)** empirical validation of the new objective show our models have state of the art generation results for idempotent models and strong zero-shot editing capabilities. We achieve more than a 41% reduction in FID compared to the SOTA IGN model.

The manuscript is organized as follows: first, we describe the probability flow ordinary differential equation and idempotent models that underpin our study. We then introduce our novel learning objectives and provide theoretical insights into them. We then contextualize our contributions with a review of related work. Finally, we provide empirical experiments showcasing state-of-the-art performance for IGN models and discuss future research directions.

## 2 BACKGROUND

Our work focuses on establishing a connection between diffusion or score-based models and IGNs to improve IGN training stability. We achieve this by training on samples along the probability flow differential equation (PF-ODE) Song et al. (2020a) defined by score-based models learning to project onto an idempotent data manifold.

**Notation**. We denote the unknown, true data distribution with $\mathcal{P}_{\text{data}}$, and the corresponding score of the probability density is defined as $\nabla_{\mathbf{x}} \log \mathcal{P}_{\text{data}}$. $\mathbf{x} \sim \mathcal{P}_{\text{data}}$ are objects sampled from the data distribution. The score function, $s_\phi(\cdot)$, is a parameterized function trained to approximate $\nabla_{\mathbf{x}} \log \mathcal{P}_{\text{data}}$. $O(\mathbf{x}, t) = \mathbf{x} \circledast \mathcal{N}(0, \sigma(t)\mathbf{I})$ is the noising operator, where $\sigma(t)$ is the noise schedule defined as a monotonically increasing function of time $t$ and $\circledast$ denotes the convolution of the two distributions. $\mathcal{P}_{\text{data}}^{\sigma(t)}(\mathbf{x}_t)$ denotes the noising operator perturbed data distribution at time $t$. Our models are trained to learn $\mathcal{P}_{\text{model}}$, under the condition that it is sufficiently similar to $\mathcal{P}_{\text{data}}$. We use $\mathcal{U}[[1, N]]$ as the uniform distribution over the set $\{1, 2, \dots, N\}$. The sequence of time is discretized with the set $\{t_i\}_{i=0}^N$, where $t_0 = \epsilon$ and $t_N = T$.

### 2.1 PROBABILITY FLOWS ODE

Following the notations from Franceschi et al. (2023a), denoising Score-based Diffusion models are represented by the following Stochastic Differential Equation (SDE):

$$dx_t = 2\sigma(t)\dot{\sigma}(t)\nabla \log \mathcal{P}_{\text{data}}^{\sigma(t)}(\mathbf{x}_t) + \sqrt{2\sigma(t)\dot{\sigma}(t)}dW_t,$$

where $\mathcal{P}_{\text{data}}^{\sigma(t)}(\mathbf{x}_t)$ denotes the Gaussian perturbed data distribution, $\sigma(t)$ is the noise schedule function that defines the noise level at time $t$, and the dot denotes a time derivative, and $W_t$ denotes the Wiener Process. An important property of this SDE is that there exists a corresponding Ordinary Differential Equation, named Probability Flow ODE (PF-ODE), whose solution trajectory has the same marginal probability distribution as the SDE. This admits a PF-ODE:

$$\frac{d\mathbf{x}}{dt} = -\sigma(t)\dot{\sigma}(t)\nabla_{\mathbf{x}} \log \mathcal{P}_{\text{data}}^{\sigma(t)}(\mathbf{x}_t). \tag{1}$$

Eq. 1 enables evolving a sample from $x_{\sigma(t_a)} \sim \mathcal{P}_{\text{data}}^{\sigma(t_a)}$ to a sample $x_{\sigma(t_b)} \sim \mathcal{P}_{\text{data}}^{\sigma(t_b)}$ (or equivalently noise scales $\sigma(t_a) \to \sigma(t_b)$). The goal of score-based generative methods is to flow samples from an easily sampleable distribution (like a Gaussian) to the true data distribution. Generally, the probability flows are constrained such that samples are reversed from time T, where $\mathcal{P}^{\sigma(T)} \approx \mathcal{N}(0, \sigma(T)\mathbf{I})$ to time $\epsilon$ where $\mathcal{P}^{\sigma(\epsilon)} \approx \mathcal{P}_{\text{data}}$. Flow-matching models Liu et al. (2022a) further generalize the mechanism of transporting samples from one distribution to another based on ODE transport.

In practice, we don't know the true $\mathcal{P}_{\text{data}}^{\sigma(t)}$, and thus the function, $s_\phi(\mathbf{x}_t, \sigma(t))$, parameterized by learnable weights $\phi$, is trained to approximate $\nabla_{\mathbf{x}} \log \mathcal{P}_{\text{data}}^{\sigma(t)}$ to obtain the empirical PF-ODE. Several ODE solving techniques Karras et al. (2022); Lu et al. (2022); Liu et al. (2022b); Zhang et al. (2022) have been proposed for the empirical PF-ODE. For example, using Euler's first-order method, the empirical PF-ODE updates can be written as,

$$\frac{\mathbf{x}_{t_n} - \mathbf{x}_{t_{n+1}}}{t_n - t_{n+1}} = -\sigma(t_{n+1})\dot{\sigma}(t_{n+1})s_\phi(\mathbf{x}_{t_{n+1}}, \sigma(t_{n+1})), \tag{2}$$

where the time horizon is partitioned into $N-1$ sub-intervals, and each time step is indexed by n. More details of the discretization step can be found in Karras et al. (2022). New samples are generated by iterating randomly sampled inputs with a score model $s_\phi$ and an ODE solver. We can see that the trained model provides us with a strong proxy for the real score function, $\nabla_{\mathbf{x}} \log \mathcal{P}_{\text{data}}^{\sigma(t)}(\mathbf{x}_t)$.

## 2.2 IDEMPOTENT GENERATIVE NETWORKS

Idempotent generative networks (IGNs) Shocher et al. (2023) are generative models based on the property of idempotence. An idempotent mapping, $f$, is an operator in some space $\mathcal{X}$ such that for some $x \in \mathcal{X}$, we have, $f(f(x)) = f(x)$. Identity and absolute value are canonical examples of idempotent operators. IGN learn a constrained idempotent mapping that is idempotent for elements of some target data manifold (e.g., real images), while projecting all other inputs to the manifold. The trained model can be used for generation by sampling from a distribution such as Gaussian noise and using the model to project the random sample to the learned data manifold. Specifically, the IGN optimization objectives rely on three main principles: **(a)** the identity boundary condition on the data manifold described above, **(b)** idempotence, and **(c)** the size of the data manifold is minimal. These objectives can be optimized by their respective loss functions.

**Reconstruction Loss**. For a sample $x \sim \mathcal{P}_{\text{data}}$, given a distance metric function $D(\cdot, \cdot)$, the reconstruction loss is:

$$\mathcal{L}_{\text{recon}} = \mathbb{E}_{\mathbf{x} \sim \mathcal{P}_{\text{data}}} [\delta_\theta(\mathbf{x})] = \mathbb{E}_{\mathbf{x} \sim \mathcal{P}_{\text{data}}} [D(\mathbf{x}, f_\theta(\mathbf{x}))]. \tag{3}$$

This imposes the boundary condition of preserving in-distribution samples on the data manifold.

**Idempotent Loss**. For any input from the domain distribution $z \sim \mathcal{Z}$, similarly with a distance metric function $D(\cdot, \cdot)$, the idempotent loss is:

$$\mathcal{L}_{\text{idem}} = \mathbb{E}_{\mathbf{z} \sim \mathcal{P}_{\mathbf{z}}} [\delta_{\theta'}(f_\theta(\mathbf{z}))] = \mathbb{E}_{\mathbf{z} \sim \mathcal{P}_{\mathbf{z}}} [D(f_\theta(\mathbf{z}), f'_\theta(f_\theta(\mathbf{z})))], \tag{4}$$

where $f'_\theta$ is the frozen copy of the model $f_\theta$. $\mathbf{z}$ is restricted to be from some easily sampleable distribution such as $\mathcal{N}(\mathbf{0}, \mathbf{I})$. This loss is minimized when for any sample, $z$, the model is idempotent: $f_\theta(f_\theta(z)) = f_\theta(z)$.

**Tightness Loss**. For any input from the domain distribution $z \sim \mathcal{Z}$, the tightness loss is:

$$\mathcal{L}_{\text{tight}} = \mathbb{E}_{\mathbf{z} \sim \mathcal{P}_{\mathbf{z}}} [\delta_\theta(f_{\theta'}(\mathbf{z}))] = \mathbb{E}_{\mathbf{z} \sim \mathcal{P}_{\mathbf{z}}} [-D(f_\theta(f'_\theta(\mathbf{z})), f'_\theta(\mathbf{z}))]. \tag{5}$$

As above, $f'_\theta$ is the same as $f_\theta$. The IGN training algorithm and gradient equations are reproduced in Appendix C for clarity and completeness. The IGN loss function is, therefore,

$$\mathcal{L}_{\text{IGN}} = \mathcal{L}_{\text{recon}} + \mathcal{L}_{\text{idem}} + \lambda_t \mathcal{L}_{\text{tight}}, \tag{6}$$

where $\lambda_t$ is the weight of the tightening loss term. The authors set $\lambda_t < 1$ to stabilize training.

The training of IGN requires all terms of Eq. 6 to be minimized simultaneously. This empirically leads to training instabilities. For one, the $\mathcal{L}_{tight}$ objective is the opposite of the $\mathcal{L}_{idem}$ objective, introducing adversarial optimization and making training more difficult. Furthermore, the minimum of $\mathcal{L}_{tight}$, without reweighting, is unbounded, which can lead to arbitrarily large gradient updates. Even with the reweighted tightening loss in Shocher et al. (2023) can produce large gradient updates when both $\mathcal{L}_{\text{tight}}$ and $\mathcal{L}_{\text{rec}}$ are large positive numbers, and lead to unstable training dynamics. Shocher et al. (2023) note IGNs have similar training characteristics to GANs, which are well known to suffer from training instability and mode collapse Kodali et al. (2017); Salimans et al. (2016); Arjovsky and Bottou (2017); Saxena and Cao (2021).

## 3 SCORE-BASED IDEMPOTENT MODEL

Score-based Idempotent Generative (SIGN) models distill pre-trained score models to improve IGN training dynamics and enable fast sampling. As previously mentioned, the adversarial tightening loss is a key cause of poor training dynamics on IGNs. Crucially, we note that the IGN objective can be achieved using the score function learned by diffusion models. Specifically, for a Diffusion or flow-matching model, we have a solution trajectory $\{\mathbf{x}_t\}_{t \in [\epsilon, T]}$ based on the corresponding PF-ODE as in Eq.1. On this trajectory, only the point $x_\epsilon$ is on the data distribution $\mathcal{P}_{\text{data}}$, while the rest of the

points on the trajectory are off the manifold and naturally form a constraint on the size of the data manifold.

The main contribution of our work is the **score-based idempotent loss**, $\mathcal{L}_{\text{SIGN}}$, for training idempotent generative models. We replace the unbounded tightening loss with a **distribution matching loss**, $\mathcal{L}_{\text{dmd}}$, or a **consistency flow loss**, $\mathcal{L}_{\text{flow}}$, to learn the data manifold from a score-function estimate for an idempotent generative model.

**Distribution Matching Loss**. The goal of a generative model is to ultimately sample from the data distribution by sufficiently emulating some target distribution. A diffusion or flow-based learns a target distribution $\mathcal{P}_{\text{target}}$ that is sufficiently close to $\mathcal{P}_{\text{data}}$. While we do not have access to either distribution, we have access to a learned approximation of $\nabla_{\mathbf{x}_t} \log \mathcal{P}_{\text{data}}^{\sigma(t)}(\mathbf{x}_t)$ through the trained diffusion model. By matching the scores of our idempotent model over a family of noisy distributions with the scores of our pre-trained model,

Following Yin et al. (2024a), we also estimate the probability densities by doing gradient updates using approximate scores. We directly

$$\nabla_\theta \mathcal{L}_{\text{DMD}} = \mathop{\mathbb{E}}_{\substack{\mathbf{z} \sim \mathcal{N}(0, \mathbf{I}), \\ n \sim \mathcal{U}[[1, N]]}} \left( s_{\text{learned}}(y_n, n) - s_{\text{diffusion}}(y_n, n) \right) \frac{df_\theta}{d\theta} \tag{7}$$

where, $\mathbf{y}_n = O(f_\theta(\mathbf{z}), t_n.)$ $O(\mathbf{x}, t)$ is the noising operator, such that, $O(\mathbf{x}, t) = \mathbf{x} \circledast \mathcal{N}(0, \sigma(t)\mathbf{I})$, which performs t steps forward diffusion on the given input $\mathbf{x}$. We use a pre-trained diffusion model for $s_{\text{diffusion}}(,)$. An auxiliary diffusion model is trained along with the SIGN model to provide the learned score estimates, $s_{\text{learned}}(,)$.

**Consistency Flow Loss**. $\mathcal{L}_{\text{DMD}}$ requires an additional model tracking the scores of the current model during training, incurring a large computation cost. As an alternative, we take inspiration from consistency models to minimize the size of the manifold through the probability flow ODE and propose the flow loss. Based on a diffusion-based ODE solver to impose restrictions on the learned manifold, the flow-based loss is given by:

$$\mathcal{L}_{\text{Flow}} = \mathop{\mathbb{E}}_{\mathbf{x} \sim \mathcal{P}_{\text{data}}, n \sim \mathcal{U}[[1, N]]} [D(f_\theta(\mathbf{x}_{t_n}), f_{\theta'}(\mathbf{x}_{t_s}))] \tag{8}$$

where, $\mathbf{x}_{t_n} = O(\mathbf{x}, t_n)$ and $\mathbf{x}_{t_s}$ is obtained by taking a step following the empirical PF-ODE in Eq. 2 using a learned score function, like a trained diffusion model, or an empirical score-estimator. The real score function, $\nabla_x \log \mathcal{P}_{\text{data}}$, defines a vector field over the data space $x$. By taking the expectation over $\mathbf{x}$ and noises $t_n$, we obtain the average direction of motion at any $\mathbf{x}$, which is used as the score function estimate. Given constraints, such as linearity in the trajectory, as shown in Theorem 3, this allows us to learn the target manifold. Intuitively, this loss can be seen as treating points along the trajectory as off-manifold domains, and we require them to be mapped in one step onto the same point of the data manifold. We use the training techniques from Consistency Models, such as using pre-trained diffusion models, or an unbiased estimator for the ODE-solvee to approximate the score function.

**Improving Training Dynamic** While the aforementioned objectives are sufficient in theory, we introduce further improvements to speed up convergence and improve generation quality. We use 2 auxiliary loss terms **(a)** regression loss and **(b)** denoising loss for this purpose. Yin et al. (2024a) show that the regression loss of supervised learning on generated pairs from a pre-trained model significantly improves model quality. Similarly, the denoising loss connects the training objective to a test-time scenario where noisy samples may be input to the model to iteratively improve generated outputs.

**Score-based Idempotent Loss**. Using the consistency loss over the modified distributions obtained by adding Gaussian noise to the data distribution, we propose the consistency-based IGN loss as:

$$\mathcal{L}_{\text{SIGN}} = \mathcal{L}_{\text{recon}} + \mathcal{L}_{\text{idem}} + \lambda_f \mathcal{L}_{\text{flow}} + \lambda_d \mathcal{L}_{\text{dmd}} + \lambda_r \mathcal{L}_{\text{reg}} + \lambda_n \mathcal{L}_{\text{denoise}} \tag{9}$$

where, $\lambda_f, \lambda_d, \lambda_r, \lambda_n$ are hyperparameters for each auxiliary loss terms. In practice, we set $\lambda_f, \lambda_d, \lambda_r, \lambda_n \in [0, 1]$, to optionally enable various loss terms. Depending on the training environment and dataset complexity, a subset of the loss terms is used. The coefficents are decided heuristically so that all terms have the same magnitude at the start of training.

We replace the unbounded tightening loss in Eq. 5 with a combination of distribution matching and flow-based losses in Eq.9. The tightening loss in the original IGN aims to restrict the manifold to only include the data distribution, but causes training instability. We directly restrict the learned data manifold toward the data distribution. By replacing the unbounded loss and directly restricting the learned data manifold, we ensure all components of our objective are bounded for stable training. In the following section, we show in Theorem 1 and Theorem 2, the distribution matching and flow-based losses accomplish the same objective. The training pseudo-code for the complete training loss and all constituent losses is in Appendix A.1.

**Sampling** SIGN aims to excel at single-step generation, which is its primary mode of operation. However, similar to Consistency models, IGN, and Diffusion models, SIGN also provides the ability to perform multi-step sampling to trade off computational cost for generation quality. Generated outputs can be iteratively refined by computing multiple forward passes on the data. The recursive sampling algorithm is presented in Alg. 2.

Karras et al. (2022) show that additional noise injected during the sampling process improves the quality of the generated images. As the signal-to-noise ratio of the initial sample is 0, the additional noise during sampling allows the model to correct imperfections during the early stages of generation. In algorithm 3, we provide an additional sampling method that effectively "pushes" the sample out of the manifold. The additional noise injection cancels out the error introduced when sampling from a low SNR region. Different use cases may be beneficial for different sampling approaches. With inputs with a high signal-to-noise ratio, such as the case for partially corrupted or low-resolution images, the recursive sampling approach may be more favorable. On the other hand, unconditional generation may benefit from backtracking and adding additional noise in the sampling procedure in Algorithm 3.

## 4 THEORETICAL ANALYSIS

In this section, we provide convergence guarantees and error bounds and build on our understanding of the proposed SIGN model.

**Theorem 1.** *Given a trained SIGN model, $f_\theta$, such that it is a measurable idempotent map, $f_\theta : \mathbb{R}^d \to \mathbb{R}^d$. Let $\mathcal{P}_{data}$ be the true data distribution and $\mathcal{P}_{f_\theta} := f_\theta \# \mathcal{P}_{data}$ its pushforward through $f_\theta$. Given the regular and k-dimensional connected, $\mathcal{C}^2$ manifolds $D$ and $M$, we have $\operatorname{supp} \mathcal{P}_{data} = D$ and $\operatorname{supp} \mathcal{P}_f = M$, and the score functions are defined on $\forall x \in D \cap M$, if $\theta$ is the global minimum $\mathcal{L}_{SIGN}$, then $D = M$.*

The proof consists of showing that the support of the learned distribution and data distribution are included in each other the densities are equal on the manifold. The full proof is included in Appendix B. In practice, a pre-trained model or an empirical score estimator is used to obtain the data score function. Score models may not cover the whole manifold, resulting in bias. Intuitively, for points not on the data manifold, the real score function has non-zero gradients, pushing the pushforward distribution towards the real distribution, contracting the manifold. Interestingly, Kamb and Ganguli (2024), Biroli et al. (2024), and De Bortoli (2022) analytically show that diffusion models recover target distributions on low-dimensional manifolds. Empirically, our method attempts to learn a mapping to this learned manifold.

Next, we look at the convergence conditions for our proposed flow-based SIGN loss.

**Theorem 2.** *Denote the distribution learned by the trained SIGN model $f_\theta$ as $\mathcal{P}_\theta$. Assuming a large enough model capacity such that:*

$$\exists \theta^* = \arg \min_\theta \mathcal{L}_{recon} = \arg \min_\theta \mathcal{L}_{flow} = 0$$

*then the learned distribution $\mathcal{P}_\theta = \mathcal{P}_{data}$, the true data distribution.*

We can see from Theorem 2 that imposing consistency across noise levels of the PF-ODE trajectories can sufficiently tighten the manifold to capture the underlying data distribution. But we must note that the above holds for a case of sufficiently large models, perfect projection at all noise levels, non-overlapping trajectories, as well as an infinitely accurate discretization of the PF-ODE. Unfortunately, in practice, such conditions are not feasible. Particularly, obtaining trajectories by reversing the

pre-trained score model is prohibitively expensive. This necessitates adding additional loss terms as described in the previous section to improve training dynamics. The quality of the estimate thus decides the applicability of the learned SIGN. In the next theorem, we show that if the learned score function can generate samples of empirical PF-ODE trajectories with errors uniformly bounded by some noise-related quantity, we can guarantee that the error of the learned SIGN to the optimal idempotent function is bounded.

**Theorem 3.** *Let $\Delta = \max |\sigma(t_{n+1}) - \sigma(t_n)|$ for $n \in \{0, N-1\}$ and $f$ be the optimal idempotent function. For some learned model $f_\theta$ which satisfies the L-Lipschitz condition. Denote $\{\mathbf{x}_t\}_{t \in [\epsilon, T]}$ the exact PF-ODE trajectory by updating using Eq. 1, and $\{\hat{\mathbf{x}}_t\}_{t \in [\epsilon, T]}$ the empirical results by Eq. 2 (i.e., using $\mathbf{x}_{t_{n+1}}$ to solve step $n$ in Eq.2 gives the resulting $\hat{\mathbf{x}}_{t_n}$). Assume the local approximation error of updating PF-ODE, $||\hat{\mathbf{x}}_{t_n} - \mathbf{x}_{t_n}||_2$, is uniformly bounded by, $\mathcal{O}((\sigma(t_{n+1}) - \sigma(t_n))^{p+1}) \forall n \in \{1, N-1\}$ with $p \geq 1$, and $L \in \mathbb{R}_{\geq 0}$, . If $\mathcal{L}_{Flow}(\theta) = 0$, and $\mathcal{L}_{Recon}(\theta) = 0$, then we have,*

$$\sup_{\mathbf{x}_{t_n}} ||f_\theta(\mathbf{x}_{t_n}) - f(\mathbf{x}_{t_n})||_2 = \mathcal{O}((\Delta)^p)$$

Similar to Song et al. (2023), we base the proof on the global error bounds for numerical ODE solvers. Due to space limitations, we present the full proofs in Appendix B. Intuitively, Theorem 3 shows an important characteristic of SIGNs and ODE trajectories. As the upper-bound error is dependent on the truncation error of the trajectories, paths with high curvature will have high error. Therefore, while the SIGN algorithm works for all diffusion, score, and flow models, algorithms with linear trajectories are better suited. This falls in line with observations in Karras et al. (2022); Liu et al. (2022a); Liu (2022); Lee et al. (2024a).

## 5 RELATED WORK

**IGNs, GANs, and Stability**. Our work focused on improving the training of IGN, a model class for fast, single-step generation. IGN assumes an underlying data manifold and proposes a transformation $f$ that maps any input source to the manifold. In addition, all the data points on the manifold have to be mapped to itself (thus, idempotent) while minimizing the size of the manifold. Jensen and Vicary (2025) present a modified backpropagation method to enforce idempotency on generative networks based on perturbation theory for MLP and CNN-based networks, but still suffer from the sensitivity to training dynamics. IGNs are similar GANs as both contain adversarial loss terms. IGNs are particularly similar to EB-GAN Zhao et al. (2016), but IGN doesn't require the input source to be a random noise. The instability of GAN-like models is well documented in literature (Durall et al., 2020; Yu et al., 2020; Zhao et al., 2018), and significant care is required to train large-scale adversarial models. Interestingly, Franceschi et al. (2023b) show a similar connection between GAN models and score models, where they train GANs using pre-trained models. We provide improved performance, additional theoretical guarantees for IGNs using score models, as well as practical training recipes.

**Diffusion, Score-based Models, and Distillation**. Diffusion models Sohl-Dickstein et al. (2015); Song et al. (2020a) produce high-quality images through a slow and iterative process, which incurs high computational costs. To mitigate this challenge, fast sampling and model distillation methods are of great interest. Consistency Models Song et al. (2023) enable single-step generation by mapping all points on a PF-ODE trajectory to a single output. We show the connection between Consistency Models and IGNs and propose a novel loss to stabilize the training of the IGN models. Furthermore, Lipman et al. (2022); Liu et al. (2022c); Lee et al. (2024b) flow matching similarly casts generative modeling as a transport mapping problem, where straight trajectories in rectified flows improve sampling efficiency. Idempotent models can similarly be distilled from flow-matching models as well.

## 6 RESULTS

We train multiple SIGN models on MNIST, CIFAR-10, and CelebA datasets to get empirical validation for our theoretical results. We also use pre-trained models to perform zero-shot editing. The training details are described in Appendix D.

**Image Generation**. We evaluated our model's image generation capabilities on three standard benchmarks: MNIST Deng (2012), CIFAR-10 Krizhevsky et al. (2009), and CelebA Liu et al. (2015).

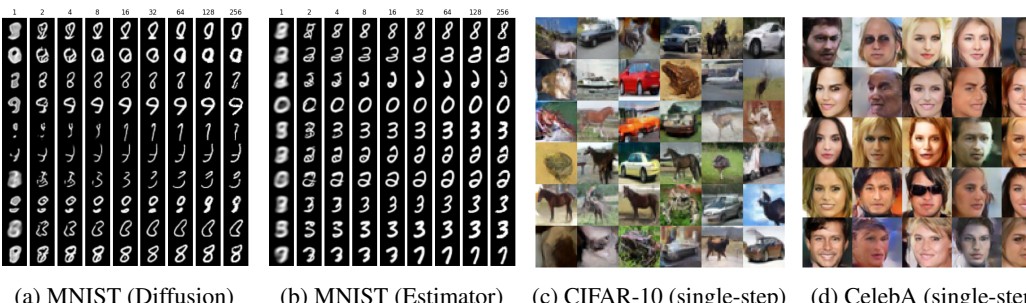

| (a) MNIST (Diffusion) | (b) MNIST (Estimator) | (c) CIFAR-10 (single-step) | (d) CelebA (single-step) |

Figure 2: Results of unconditional sampling from SIGN models. (a) Single and Multi-step generation of a SIGN model trained on MNIST using a diffusion model as the score function. (b) Single and Multi-step generation using an empirical score estimator as the score function. (c) Random sampling of single-step generation from a SIGN model trained on CIFAR-10. (d) Random sampling of single-step generation from a SIGN model on CelebA.

These datasets consist of 28x28 grayscale images, 32x32 color images, and 64x64 color images, respectively.

For MNIST, we used a convolutional U-Net as the model with and without time embeddings for the pre-trained diffusion model and the distilled SIGN model, respectively. We used a $\ell_2$ distance as our distance metric. We use a pre-trained diffusion model and an unbiased score estimator to train our SIGN models for MNIST. For the simple dataset, we do not use distribution matching, regression, or denoising loss. As illustrated in Fig. 2a and Fig. 2b, our SIGN models are capable of unconditional generation in a single pass, and image quality is progressively enhanced with multiple passes.

For CIFAR-10 and CelebA, we adapted model structures from EDM Karras et al. (2022) and DDIM Song et al. (2022) to make use of their pre-trained weights as a starting point for our training. We show the result of single-step unconditional generation in Fig. 2c and Fig. 2d. Following Alg. 3 and 2, we iteratively sample to generate samples. As the samples are more challenging, we use an additional regression loss term to stabilize the training. To evaluate performance, we first generated 50,000 unconditional, single-step samples from our trained model for each dataset, then we calculated the Fréchet Inception Distance (FID) using 2048 features to measure the similarity of the generated samples to the target dataset. Our model sets a new state-of-the-art benchmark for idempotent models by achieving FID scores of 11.09 on CIFAR-10 and 23.32 on CelebA. Our CelebA FID score significantly outperforms the original IGN model (FID=39). Prior work on idempotent models in Shocher et al. (2023) and Jensen and Vicary (2025) do not train on CIFAR-10; as such, we are the first to report CFIAR-10 results for idempotent models. For our initial ablation study, we trained a model on the CelebA dataset for the same 350 epochs but without our proposed loss. The model achieved an FID score of 123.70, which indicates that our proposed loss significantly improves the training dynamics.

**Zero-shot Editing**. We investigated the zero-shot image editing capabilities of our SIGN models on the CelebA and CIFAR-10 datasets. As shown in Fig. 3b and Fig. 3a, we first applied a checkerboard binary mask to corrupt the data, and tested how the model would perform in single-step and multi-step scenarios. For multi-step sampling, we applied a customized 10-step noise schedule. The model was able to project the corrupted image back towards the target data manifold from single-step sampling, despite not being specifically trained for this task. The multi-step results further improve the image quality, yielding a result closer to the original. We should point out that because CelebA has a higher resolution than CIFAR-10, the defects would be less obvious in Fig. 3b than in Fig. 3a.

## 7 LIMITATIONS

Though not competitive with general SOTA generative models, we perform better than current IGN models. We show that IGN models can be stably trained using non-adverserial losses. As Jensen and Vicary (2025) show, idempotent networks are a nascent line of generative models, and there is significant room for improvement in the inductive biases and training methods required to optimize these models. Intuitively, using a single network to learn both a large projection mapping from

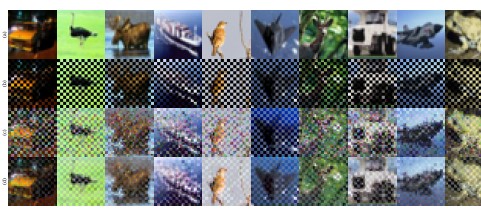 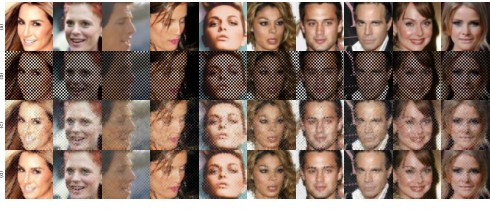

(a) CIFAR-10.        (b) CelebA

Figure 3: Zero-shot masked image editing with a SIGN model trained. From *top* to *bottom*: (1) Original images. (2) Masked inputs. (3) Single-step sampling results. (4) Multi-step sampling results.

noise to the data manifold, while also learning identity mapping on the manifold, is difficult, as the objective is significantly different from the usual generative modeling tasks. As a result, we cannot take advantage of the standard practices of generative models and the wealth of accumulated knowledge from the usual generative modeling community. We hypothesize, transformer-based and especially mixture-of-experts-based Chen et al., 2022; Shazeer et al., 2017; Riquelme et al., 2021 networks may provide significantly improved performance and bring IGNs on par with other single-step and distilled models. We plan on exploring architectural choices to improve model capabilities along with larger models, higher training compute budgets, and larger, modern datasets. Furthermore, our work greatly improves the training stability of idempotent generative models, which is a key requirement in enabling future work on high-resolution datasets. The regression loss requires pre-generation of a large set of images, requiring a large amount of compute and memory. However, the regression loss is not required, as Yin et al. (2024b) shows that regression loss is not required to obtain strong generation results. We also acknowledge that our initial ablation study is limited due to computational resource constraints, but we plan to further investigate it when resources permit.

## 8 FUTURE WORK

A key feature of our work is the improved training dynamics of idempotent generative models by utilizing learned score functions. Due to training instability in the adversarial loss, prior work on IGNs is difficult to reproduce and focused on simpler datasets. Optimizing modeling choices and training method is challenging when training is unstable. We hope our work will enable large-scale architectural optimization studies for idempotent networks that may be transferred to conventional IGN training as well. Furthermore, the inductive bias of idempotent networks is a natural fit for many other learning and generative tasks, especially in scientific workloads.

SIGN models can be combined with existing score-based models in two ways: either by "fast-forwarding" the reverse process by inputting partially denoised samples to the SIGN generator, similar to denoising diffusion GANs Xiao et al. (2021), or by employing multi-step iterative sampling schemes inspired by Shocher et al. (2023) and Song et al. (2023). As noted in Shocher et al. (2023), the model may also benefit from a two-step approach as in Rombach et al. (2022) instead of directly applying to the pixel-space. Furthermore, flow-matching methods such as rectified flows Liu et al. (2022c) provide an attractive alternative to diffusion models as teacher models due to their straight trajectories.

## 9 CONCLUSION

In this work, we connect Idempotent generative models with score-based diffusion models. Our proposed new losses to train IGN models improve the training characteristics of IGNs and provide theoretical guarantees of our optimization methods, and strong empirical results for idempotent models. We provide a first baseline on CIFAR10, as well as improving SOTA CelebA FID by more than 40% for idempotent models. We view this work as an initial step towards connecting IGNs with score-based generative models that allow the development of more powerful models. Connecting to score-based opens the door for transferring learning techniques, model architecture, and even learned weights to improve IGN models. A more stable training algorithm can enable further exploration into identifying network architectures that are suitable for idempotent models and allow learning on large-scale, higher-resolution datasets. We plan on exploring such possibilities in the future.

## 10   REPRODUCIBILITY STATEMENT

The complete proofs of all theorems in our work, discussed in Section 4, are detailed in Appendix B. The source code used in our experiments and details regarding how to reproduce our experiments can be found in Appendix D.

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

## A ALGORITHM

### A.1 TRAINING ALGORITHMS

---

**Algorithm 1** Consistent Idempotent Training

---

1: **Input:** Dataset $\mathcal{D}$, models $f_\theta$, $f_{\theta'}$, distance metric, $D(\cdot, \cdot)$, noising operator $O(\cdot, \cdot)$, learning rate $\eta$, loss term hyperparameters $\lambda_f, \lambda_d, \lambda_r, \lambda_n$
2: **while** not converged **do**
3:     Copy $f_{\theta'} \leftarrow f_\theta$
4:     Sample $\mathbf{x} \sim \mathcal{D}$
5:     Sample $\mathbf{z} \sim \mathcal{N}(\mathbf{0}, \mathbf{I})$
6:     Sample $n \sim \mathcal{U}([[1, T]])$
7:     $\mathbf{x}_{t_n} \leftarrow O(\mathbf{x}, t_n)$
8:     Obtain $\mathbf{x}_{t_s}$ from solving steps in Eq. 2.
9:     $\mathbf{x}_{\text{recon}}, \mathbf{x}_{\text{sample}} \leftarrow f_\theta(\mathbf{x}), f_\theta(\mathbf{z})$
10:     $\mathbf{x}_{\text{idem}} \leftarrow f_\theta(f_{\theta'}(\mathbf{z}))$
11:     Copy $\mathbf{x}_{\text{clone}} \leftarrow x_{\text{sample}}$
12:     $y_n \leftarrow O(f_\theta(\mathbf{x}_{\text{sample}}), t_n.)$
13:     $\mathcal{L}_{\text{recon}} \leftarrow D(\mathbf{x}, \mathbf{x}_{\text{recon}})$
14:     $\mathcal{L}_{\text{idem}} \leftarrow D(\mathbf{x}_{\text{sample}}, \mathbf{x}_{\text{idem}})$
15:     $\mathcal{L}_{\text{denoise}} \leftarrow D(\mathbf{x}, f_\theta(\mathbf{x}_{t_n}))$
16:     $\mathcal{L}_{\text{flow}} \leftarrow D(f_\theta(\mathbf{x}_{t_n}), f_{\theta'}(\mathbf{x}_{t_s}))$
17:     $\mathcal{L}_{\text{DMD}} \leftarrow D(s_{\text{learned}}(y_n, n) - s_{\text{diffusion}}(y_n, n))$
18:     $\mathcal{L} \leftarrow \mathcal{L}_{\text{recon}} + \mathcal{L}_{\text{idem}} + \lambda_f \mathcal{L}_{\text{flow}} + \lambda_d \mathcal{L}_{\text{dmd}} + \lambda_r \mathcal{L}_{\text{reg}} + \lambda_n \mathcal{L}_{\text{denoise}}$
19:     $f_\theta \leftarrow f_\theta - \eta \nabla_\theta \mathcal{L}(f_\theta)$
20: **end while**

---

The learned diffusion score-model $s_{\text{learned}}$ is trained online as in Yin et al. (2024a).

---

**Algorithm 2** Recursive Sampling

---

1: **Input:** Trained CIGN $\mathbf{f}_\theta(\cdot)$, initial noise $\mathbf{x}_T$
2: $\mathbf{x} \leftarrow \mathbf{f}_\theta(\mathbf{x})$
3: **while** not converged **do**
4:     $\mathbf{x} \leftarrow \mathbf{f}_\theta(\mathbf{x})$
5: **end while**

---

**Algorithm 3** Multistep Sampling with Editing

---

1: **Input:** Trained CIGN $\mathbf{f}_\theta(\cdot)$, initial noised data $\mathbf{x}'$, image mask $\mathbf{M}$
2: Noise schedule $0 < \sigma_N < \sigma_{N-1} \ldots < \sigma_1$
3: $\mathbf{x} \leftarrow \mathbf{f}_\theta(\mathbf{x}') \odot M + \mathbf{x}' \odot (1 - M)$
4: **for** $i = 1, i < N$ **do**
5:     Sample $\mathbf{z} \sim \mathcal{N}(\mathbf{0}, \mathbf{I})$
6:     $\mathbf{x}_\tau \leftarrow \mathbf{x} + \sigma_i z$
7:     $\mathbf{x} \leftarrow \mathbf{f}_\theta(\mathbf{x}_\tau) \odot M + \mathbf{x} \odot (1 - M)$
8: **end for**

---

## B PROOFS

**Theorem.** *Given a trained SIGN model, $f_\theta$, such that it is a measurable idempotent map, $f_\theta : \mathbb{R}^d \rightarrow \mathbb{R}^d$. Let $\mathcal{P}_{data}$ be the true data distribution and $\mathcal{P}_{f_\theta} := f_\theta \# \mathcal{P}_{data}$ its pushforward through $f_\theta$. Given the regular and connected manifolds $D$ and $M$, we have $\text{supp}\,\mathcal{P}_{data} = D$ and $\text{supp}\,\mathcal{P}_f = M$, and the score functions are defined on $\forall x \in D \cap M$, if $\theta$ is the global minimum $\mathcal{L}_{SIGN}$, then $D = M$.*

*Proof.* We start with our trained idempotent model $f_\theta$ and the definition of the manifold $M$. As the $\theta$ is the global minimizer of $\mathcal{L}_{\text{SIGN}}$, we have a perfectly idempotent model. We have $f_\theta : \mathbb{R}^d \to \mathbb{R}^d$, and

$$M := \text{Im}(f) = \{y : \exists x, y = f_\theta(x)\}$$

where every $y \in M$ is a fixed point such that $f_\theta(y) = y$. We have a learned distribution distribution $p_{f_\theta} := f_\theta \# \mathcal{P}_{\text{data}}$. We also have $\text{supp} \, \mathcal{P}_{\text{data}} = D$ and $\text{supp} \, \mathcal{P}_f = M$. Finally, since we have that $\theta$ is the global minimizer of $\mathcal{L}_{\text{SIGN}}$, crucially $\mathcal{L}_{\text{DMD}}$ is minimized. Minimizing the distribution score matching loss results in a score equality over the support of distributions.

On each $C^2$ manifold, with volume measures $\mu_M$ and $\mu_D$, let's define positive densities, $q_M$ and $q_D$ with respect to the volume measures, $\mu_M$ and $\mu_D$.

Therefore, the tangential scores of each manifold-density are equal:

$$\nabla_T \log q_M = \nabla_T \log q_D \quad \forall x \in D \cap M$$

Since the densities must be normalized, the normalization constant is 0, and they must be equal. Assume for a contradiction that there exists $x_o \in D$ and $x_0 \notin M$. Thus, there is an open neighborhood $U$ of $x_0$ where $\mathcal{P}_{\text{data}}$ has a positive manifold density $q_D > 0$. But since $x_0 \notin M$, $q_M \leq 0$ or does not exist in $U$. This contradicts the equality of manifold densities; thus, there is no $x_0$ such that $x_0 \in D$ but $x_0 \notin M$. Thus, we have $\text{supp}(\mathcal{P}_{\text{data}}) \subseteq \text{supp}(\mathcal{P}_{f_\theta})$ and equivalently, $D \subseteq M$.

Minimizing $\mathcal{L}_{\text{SIGN}}$ also requires minimizing the $\mathcal{L}_{idem}$. As $\mathcal{P}_{f_\theta}$ is the idempotent pushforward of $\mathcal{P}_{\text{data}}$, we have $\text{supp} \, \mathcal{P}_{f_\theta} \subseteq f_\theta(\text{supp} \, \mathcal{P}_{\text{data}})$. Since, $\text{supp} \, \mathcal{P}_{\text{data}} \subseteq \text{supp} \, \mathcal{P}_{f_\theta}$, we have, $f_\theta(\text{supp} \, \mathcal{P}_{\text{data}}) \subseteq f_\theta(\text{supp} \, \mathcal{P}_{f_\theta})$. From idempotent symmetry over the support, and have $f_\theta(\text{supp} \, \mathcal{P}_{f_\theta}) = \text{supp} \, \mathcal{P}_{f_\theta}$ and we have $\text{supp} \, \mathcal{P}_{f_\theta} \subseteq \text{supp} \, \mathcal{P}_{\text{data}}$. Thus, we have $\text{supp}(\mathcal{P}_{f_\theta}) \subseteq \text{supp}(\mathcal{P}_{\text{data}})$ and equivalently, $M \subseteq D$.

Combining, $M \subseteq D$ and $D \subseteq M$, we van see that $D = M$ and $\mathcal{P}_{f_\theta} = \mathcal{P}_{\text{data}}$, completing the proof. $\square$

**Theorem.** *Denote the distribution learned by the trained SIGN model $f_\theta$ as $\mathcal{P}_\theta$. Assuming a large enough model capacity such that:*

$$\exists \theta^* = \arg \min_\theta \mathcal{L}_{recon} = \arg \min_\theta \mathcal{L}_{flow} = 0$$

*then the learned distribution $\mathcal{P}_\theta = \mathcal{P}_{data}$, the true data distribution.*

*Proof.* Assuming the set of parameters $\theta^*$, the model $f_{\theta^*}$ minimizes the proposed flow loss, in Eq. (8), and thus we have,

$$d(f_{\theta^*}(\mathbf{x}_{t_{n+1}}), f_{\theta^*}(\mathbf{x}_{t_n})) = 0,$$

where $n \in [\epsilon, T-1]$ denotes trajectory along the PF-ODE with different noise steps. By the definition of a metric function, we have,

$$f_{\theta^*}(\mathbf{x}_{t_{n+1}}) = f_{\theta^*}(\mathbf{x}_{t_n}). \tag{10}$$

Now, let's consider the base case, $n = 0$ and $t_0 = \epsilon$. We have:

$$d(f_{\theta^*}(\mathbf{x}_{t_1}), f_{\theta^*}(\mathbf{x}_\epsilon)) = 0,$$

$$d(f_{\theta^*}(\mathbf{x}_{t_1}), \mathbf{x}_\epsilon) \overset{(a)}{=} 0,$$

$$f_{\theta^*}(\mathbf{x}_{t_1}) \overset{(b)}{=} \mathbf{x}_\epsilon \tag{11}$$

where (a) is due to $f_{\theta^*}$ minimizing the reconstruction loss, therefore, $\forall \mathbf{x}_\epsilon, f_{\theta'}(\mathbf{x}_\epsilon) = \mathbf{x}_\epsilon$ and (b) is due to the definition of the distance metric. By Eq. (10), Eq. (11), and mathematical induction, we will have $f_{\theta^*}(\mathbf{x}_T) = \mathbf{x}_\epsilon$. In other words, for all random noise sampled from the source $\mathbf{x}_T \sim \mathcal{Z}$, after applying the learned CIGN transformation $f_{\theta^*}$, will fall in the terminal distribution $\mathbf{x}_\epsilon \sim \mathcal{P}^\epsilon$, which is the data distribution, $\mathcal{P}_{\text{data}}$. $\square$

**Theorem.** *Let $\Delta = \max |\sigma(t_{n+1}) - \sigma(t_n)|$ for $n \in \{0, N-1\}$ and $f$ be the optimal idempotent function. For some learned model $f_\theta$ which satisfies the L-Lipschitz condition. Denote $\{\mathbf{x}_t\}_{t \in [\epsilon, T]}$ the exact PF-ODE trajectory by updating using Eq. 1, and $\{\hat{\mathbf{x}}_t\}_{t \in [\epsilon, T]}$ the empirical results by Eq. 2 (i.e., using $\mathbf{x}_{t_{n+1}}$ to solve step $n$ in Eq.2 gives the resulting $\hat{\mathbf{x}}_{t_n}$). Assume the local approximation*

*error of updating PF-ODE, $||\hat{\mathbf{x}}_{t_n} - \mathbf{x}_{t_n}||_2$, is uniformly bounded by, $\mathcal{O}((\sigma(t_{n+1}) - \sigma(t_n))^{p+1}) \forall n \in \{1, N-1\}$ with $p \geq 1$, and $L \in \mathbb{R}_{\geq 0}$, . If $\mathcal{L}_{Flow}(\theta) = 0$, and $\mathcal{L}_{Recon}(\theta) = 0$, then we have,*

$$\sup_{\mathbf{x}_{t_n}} ||f_\theta(\mathbf{x}_{t_n}) - f(\mathbf{x}_{t_n})||_2 = \mathcal{O}((\Delta)^p)$$

*Proof.* Recall the $\mathcal{L}_{\text{Flow}}$:

$$\mathcal{L}_{\text{Flow}} = \mathbb{E}_{\mathbf{x} \sim \mathcal{P}_{\text{data}}, n \sim \mathcal{U}[[1,N]]}[D(f_\theta(\mathbf{x}_{t_n}), f_{\theta'}(\mathbf{x}_{t_s}))].$$

$\mathcal{L}_{\text{Flow}} = 0$ implies $D(f_\theta(\mathbf{x}_{t_{n+1}}), f_\theta(\hat{\mathbf{x}}_{t_n})) = 0$ and thus $f_\theta(\mathbf{x}_{t_{n+1}}) = f_\theta(\hat{\mathbf{x}}_{t_n})$. Since $f$ is the optimal CIGN solution, we have $f(\mathbf{x}_{t_{n+1}}) = f(\mathbf{x}_{t_n})$. Denote error at noise level $n$ as $\mathbf{e}_n = f_\theta(\mathbf{x}_{t_n}) - f(\mathbf{x}_{t_n})$.

We now form the recursion relation,

$$\mathbf{e}_{n+1} = f_\theta(\mathbf{x}_{t_{n+1}}) - f(\mathbf{x}_{t_{n+1}}) \tag{12}$$

$$= f_\theta(\hat{\mathbf{x}}_{t_n}) - f(\mathbf{x}_{t_n}) \tag{13}$$

$$= f_\theta(\hat{\mathbf{x}}_{t_n}) - f_\theta(\mathbf{x}_{t_n}) + f_\theta(\mathbf{x}_{t_n}) - f(\mathbf{x}_{t_n}) \tag{14}$$

$$= f_\theta(\hat{\mathbf{x}}_{t_n}) - f_\theta(\mathbf{x}_{t_n}) + \mathbf{e}_n. \tag{15}$$

Due to the Lipschitz condition, we have

$$||f_\theta(\hat{\mathbf{x}}_{t_n}) - f_\theta(\mathbf{x}_{t_n})||_2 \leq L||\hat{\mathbf{x}}_{t_n} - \mathbf{x}_{t_n}||_2.$$

Thus, we can bound the error at noise-scale $n+1$ with,

$$||\mathbf{e}_{n+1}||_2 \leq ||\mathbf{e}_n||_2 + L||\hat{\mathbf{x}}_{t_n} - \mathbf{x}_{t_n}||_2$$

Furthermore, as the local approximation error, $||\hat{\mathbf{x}}_{t_n} - \mathbf{x}_{t_n}||_2$, is uniformly bounded by, $\mathcal{O}((\sigma(t_{n+1}) - \sigma(t_n))^{p+1})$, and $L \in \mathbb{R}_{\geq 0}$, we have

$$||\mathbf{e}_{n+1}||_2 \leq ||\mathbf{e}_n||_2 + \mathcal{O}(L(\sigma(n+1) - \sigma(n))^{p+1})$$

For the base case of $\mathbf{e}_\epsilon$, $f_\theta(\mathbf{x}_\epsilon) = \mathbf{x}_\epsilon$ as we assume $\mathcal{L}_{\text{Recon}}(\theta) = 0$ and by definition, $f(\mathbf{x}_\epsilon) = \mathbf{x}_\epsilon$, and thus we have $\mathbf{e}_\epsilon = 0$.

We can now bound the error $||\mathbf{e}_n||_2$, by induction on the error of previous noise levels,

$$||\mathbf{e}_n||_2 \leq L||\hat{\mathbf{x}}_{t_{n-1}} - \mathbf{x}_{t_{n-1}}||_2 + ||\mathbf{e}_{n-1}||_2 \tag{16}$$

$$= \sum_{i=\epsilon}^{n-1} L\mathcal{O}((\sigma(t_{i+1}) - \sigma(t_i)^{p+1}) \tag{17}$$

$$\leq \sum_{i=\epsilon}^{n-1} (\sigma(t_{i+1}) - \sigma(t_i))(L\mathcal{O}(\Delta)^p). \tag{18}$$

$$= (L\mathcal{O}(\Delta)^p)(t_n - \epsilon) \tag{19}$$

As $t_n - \epsilon \leq t_N - \epsilon \leq C$, where C is some constant. We therefore have,

$$(L\mathcal{O}(\Delta)^p)(t_n - \epsilon) \leq C(\mathcal{O}(\Delta)^p). \tag{20}$$

As $C$ and $L$ are constants and can be neglected compared to the exponential term, we have $C(L\mathcal{O}(\Delta)^p) = (\mathcal{O}(\Delta)^p)$, which completes the proof. □

## C  IGN TRAINING

To be self-consistent, in Alg. 4 we reproduce the training procedure for a standard IGN in pseudocode.

## D  EXPERIMENT DETAILS

**Model Architecture**  Intuitively, we attempt to parameterize the model based on existing work on the consistency and diffusion models. We based our model on the same architecture as the pre-trained model, with our custom loss functions employed.

---

**Algorithm 4** Training an idempotent generative network

---

1: **Input:** Data set $\mathcal{D}$, models $\phi_\theta$, $\phi_{\theta^\backprime}$, drift measure $\delta(\cdot, \cdot)$
2: **while** not converged **do**
3:      Sample $x \sim \mathcal{D}$
4:      Sample $z \sim \mathcal{N}(\mathbf{0}, \mathbf{I})$
5:      Copy $\phi_{\theta^\backprime} \leftarrow \phi_\theta$
6:      $x_{\text{recon}}, x_{\text{sample}} \leftarrow \phi_\theta(x), \phi_\theta(z)$
7:      $x_{\text{idem}} \leftarrow \phi_{\theta^\backprime}(z)$
8:      $x_{\text{tight}} \leftarrow \phi_{\theta^\backprime}(z)$
9:      Copy $x_{\text{clone}} \leftarrow x_{\text{sample}}$
10:      $x_{\text{proj}} \leftarrow \phi_\theta(x_{\text{clone}})$
11:      $\mathcal{L}_{\text{recon}} \leftarrow \delta(x, x_{\text{recon}})$
12:      $\mathcal{L}_{\text{idem}} \leftarrow \delta(x_{\text{sample}}, x_{\text{idem}})$
13:      $\mathcal{L}_{\text{tight}} \leftarrow -\delta(x_{\text{proj}}, x_{\text{clone}})$
14:      $\mathcal{L} \leftarrow \mathcal{L}_{\text{recon}} + \lambda_i \mathcal{L}_{\text{idem}} + \lambda_t \mathcal{L}_{\text{tight}}$
15:      $\phi_\theta \leftarrow \phi_\theta - \eta \nabla_\theta(\phi_\theta)$
16: **end while**

---

**Training details**    We trained them on a system with 4 Nvidia H100 GPUs, using PyTorch as the framework. Since the SIGN contains a subset of the parameters of the diffusion model, we initialize the SIGN using the parameters of a trained diffusion model. Unless otherwise specified, all hyperparameters were identical to those of the respective base models. We use the original noise schedule from EDM and DDIM respectively to train our SIGN models. We also employed techniques from Distribution Matching DistillationYin et al. (2024a) of using a pre-generated image to help our model get to the target manifold faster. For CIFAR-10, we started from Karras et al. (2022), with 200K pre-generated samples and trained for 200 epochs. For CelebA, we based our work on Song et al. (2020b), with 500K pre-generated samples and trained for 350 epochs. To further ensure training stability, we initialized our models with pre-trained weights.

**Evaluation Setup**    For each dataset, we generated 50K unconditioned single-step samples from our trained model and used FID to evaluate their overall likeliness to the target dataset.

**Source Code**    The source code and instructions to run the experiment can be acquired through this link: https://anonymous.4open.science/r/SIGN-88/

