# OpenReview forum: "Score-based Idempotent Distillation of Diffusion Models"
_ICLR.cc/2026/Conference — ICLR 2026 Conference Withdrawn Submission_

### Official Review · Reviewer_gEWL · 2025-10-28

**Soundness:** 3
**Presentation:** 2
**Contribution:** 3
**Rating:** 6
**Confidence:** 3

**Summary:**

This paper proposes a new method for training idempotent generative networks (IGNs) using a distilled pre-trained score-based diffusion model. This new approach, called SIGNs, provides improved training stability over conventional adversarially-trained IGNs, The authors provide a theoretical analysis of their score-based training approach, along with experimental results showing state-of-the-art generation results for their approach as compared to conventional IGNs.

**Strengths:**

* The paper provides a good review of the background on probability flow ODEs and IGNs.
* The contributions appear to be novel and technically sound.
* The theoretical results appear to show the utility of the proposed SIGN approach, and the ability of this approach to approximate the true data distribution.
* The experimental results appear to show that SIGN significantly improves the training dynamics of IGNs, resulting in higher quality sample generation and much improved FID scores.

**Weaknesses:**

* Regarding experimental results on improved training stability, it would be helpful to include plots showing improved behavior for training SIGNs compared to IGNs, such as the training loss over time.
* An analysis of the compute runtime complexity for the proposed SIGN algorithms should be provided in the paper. How computationally expensive are the proposed SIGN training and sampling algorithms?
* Regarding the consistency-based loss in Eq. 9, some experimental results using ablations of this loss should be provided, to show the contribution of each term in this loss to training stability and the quality of samples drawn from the SIGN.
* A few sentences throughout the paper are incomplete, with significant portions of each sentence missing, such as on line 110.

**Questions:**

Revisions to this paper to answer the bulk of the questions and missing experimental results, described under “weaknesses” above, should be provided.

---

### Official Review · Reviewer_zrM3 · 2025-10-29

**Soundness:** 2
**Presentation:** 3
**Contribution:** 1
**Rating:** 2
**Confidence:** 4

**Summary:**

This paper studies Idempotent Generative Networks (IGNs). They propose a new training method that improves performance of the model and avoids training instabilities. This new training method for IGNs is based on distillation methods from diffusion/flow models. The authors demonstrate improved empirical performance for such IGNs.

**Strengths:**

* Demonstrates that consistency losses or distribution matching losses can be incorporated in IGNs, and furthermore that they improve IGNs' performance.

**Weaknesses:**

* There is no novelty, since the new losses used for IGN are directly taken from existing works in the diffusion/flow litterature. The final model is trained on a set of losses for IGNs + consistency or DMD loss.


* The empirical performance of the models are weak. IGNs reach 11 of FID on CIFAR-10, while recent consistency models are below 3 of FID.

* While incorporating consistency/DMD losses improve IGNs, the final models are still lagging behind 1-step methods that directly distill the ODE, without using the IGN framework. Thus, the question is: what is the interest of the IGN framework?  Why would one use IGNs, instead of using directly consistency models (or other types of distillation methods)?

**Questions:**

NA

---

### Official Review · Reviewer_TPNR · 2025-11-01

**Soundness:** 2
**Presentation:** 3
**Contribution:** 3
**Rating:** 2
**Confidence:** 2

**Summary:**

This paper proposes a new method for training Idempotent Generative Networks (IGNs), which the authors call Score-based Idempotent Generative Networks (SIGNs). Conventional IGNs are a recent class of generative models that support flexible single- and multi-step generation but, like GANs, suffer from unstable adversarial training and mode collapse. Diffusion models, in contrast, offer stable training and high-quality samples but are computationally expensive due to their slow, iterative sampling process.

Inspired by diffusion model, the authors proposed SIGNs to "unite" these two model families by training an idempotent model using score-based objectives, thereby eliminating the need for adversarial losses, which makes IGNs' training instability and mode collapse. SIGNs's loss replaces the unstable and unbounded $\mathcal{L}\_{tight}$ with stable, diffusion-model-based $\mathcal{L}\_{flow}$, $\mathcal{L}\_{dmd}$, $\mathcal{L}\_{reg}$, and $\mathcal{L}\_{denoise}$.

**Strengths:**

1. The motivation to eliminate the unbounded and unstable loss from IGN is intuitive.
1. The proposed stable, non-adversarial training objective $\mathcal{L}_{SIGN}$ is novel and makes makes the entire model class significantly more practical and powerful.
1. The authors well justify the SIGN with theoertically support and emprically experiments, and the experimental results show better performance than current IGN models.

**Weaknesses:**

1. During training, SIGNs require to inference a pre-trained diffusion model $s_\text{diffusion}(\cdot,\cdot)$ and to train a diffusion model $s_\text{learned}(\cdot,\cdot)$, which means its training process requires more computational resources and its performance and quality highly depend on the pre-trained diffusion model.
1. The Score-based Idempotent Loss introduce too many (4) hyperparamters, which makes the loss hard to tune for better performance. And I cannot see ablation study on how these hyperparameters are picked.
1. Idempotent models is still on its early state, thus it can only be validated on small scale of data like CIFAR-10 and CelebA, but still it'd be better to test on ImageNet like DiT.
1. There are two symbols of the copied model $f_{\theta'}(\cdot)$ and $f_{\theta}'(\cdot)$, which are confusing.
1. The loss subscripts are different in Eq (9) and in Eq (7), and (8).
1. There is no table help readers to intuitively compare the FIDs.
1. Missing ablation studies or experiments to show how the SIGN loss help reduce the unstable adversarial training and mode collapse.

**Questions:**

1. Since this model require a pre-trained diffusion model, is there any performance and latency comparision between the used pre-trained diffusion model and the trained IGN?
1. Is it possible to instead of requiring a pre-trained diffusion model and training one more diffusion model, but to mimic the training process of diffusion model to directly help train IGNs?
1. Any multi-step generated images from SIGNs? Are they better than single-step generated images from SIGNs and multi-step generated images from other IGNs?
1. The generated images are only in shape 64x64, what is the bottleneck not to try on larger images?
1. What are the advatages of IGNs or SIGNs compared to those few-step diffusion models like Flux[schnell], SANA-sprint, etc? How IGNs and SIGNs are more promising?

---

### Official Review · Reviewer_rb7p · 2025-11-03

**Soundness:** 2
**Presentation:** 1
**Contribution:** 1
**Rating:** 2
**Confidence:** 4

**Summary:**

This paper introduces Score-based Idempotent Generative Networks (SIGNs), a new framework that unifies idempotent generative models (IGNs) and diffusion models. By distilling score functions from pre-trained diffusion models, SIGN replaces the unstable adversarial tightening loss in IGNs with a stable, bounded objective that combines score-matching, distribution-matching, and flow-consistency terms.

**Strengths:**

It is very interesting to see an adoption of diffusion models to IGN.

**Weaknesses:**

W1. Overly complex and unmotivated loss design (Eq. 9).

- The proposed objective combines 6 different loss terms, but the paper does not justify this design through ablation or theoretical analysis. A well-structured paper should demonstrate why each component is necessary. Here, the design philosophy feels more like “include every possibly helpful loss” rather than a principled choice. Unlike canonical models such as VAE, GAN, NF, or diffusion, each of which has a single interpretable loss with analyzable dynamics, this composite formulation lacks understanding of how each part contributes or interacts (especially when scaled). Unless the performance gain is overwhelmingly strong (which is not) the current design philosophy weakens the paper’s coherence and scientific clarity.

W2. Ambiguous goal and framing.

- The paper presents itself as an IGN but functionally behaves more like a distillation or fine-tuning method for pre-trained diffusion models. If the true goal is efficient diffusion model distillation, the contribution should be reframed as a distillation work, emphasizing how idempotency emerges as a by-product of the distillation loss. In its current form, the positioning between “idempotent modeling” and “score distillation” is conceptually unclear.

W3. Insufficient empirical evaluation and missing tables.

- The work lacks clear quantitative summaries, e.g., no comparison tables against baselines, no ablation results showing the contribution of each loss, and no scaling or control experiments to substantiate claims of stability or efficiency. A concise table comparing SIGN with prior IGNs, diffusion distillation, and consistency models on FID and runtime would make the results more convincing. If computational limits prevent full scaling, smaller controlled studies should still be reported to justify design choices.

**Questions:**

Between lines 157 and 163, the paper describes Karras’s work as if it provided a detailed explanation of the Euler solver, but Karras actually introduced the Heun solver, not Euler. The correct citation here should likely be DDIM rather than EDM. For discussion of DDIM and Euler solvers, see reference [1].

The notation is highly inconsistent. For instance, using $x$ where $\mathbf{x}$ should appear. Such inconsistency makes the paper appear unpolished and not yet ready for publication. Attention to precise notation is a mark of professional scientific writing; it is separate from generating good ideas but equally essential. Top-tier conferences have little tolerance for sloppy presentation.

Finally, the paper’s notation differs from conventions widely used in the community, forcing readers to spend unnecessary time translating symbols. The authors should align their notation with established community standards for clarity and accessibility.

[1] Lai, Chieh-Hsin, et al. "The Principles of Diffusion Models." arXiv preprint arXiv:2510.21890 (2025).

**Details Of Ethics Concerns:**

No conerns

---

### Note · Authors · 2025-12-05

I have read and agree with the venue's withdrawal policy on behalf of myself and my co-authors.